# Comparative Analyses of Gene and Protein Expressions and the Lipid Contents in Intramuscular and Subcutaneous Fat Tissues in Fattening Steers

**DOI:** 10.3390/ani15182733

**Published:** 2025-09-19

**Authors:** Kaixi Ji, Ming Yang, Ziying Tan, Hongbo Zhao, Xianglun Zhang

**Affiliations:** 1Shandong Province Key Laboratory of Animal Microecologics and Efficient Production of Livestock and Poultry, Key Laboratory of Livestock and Poultry Multi-Omics of MARA, Institute of Animal Science and Veterinary Medicine, Shandong Academy of Agricultural Sciences, Jinan 250100, China; jikaixi@nieer.ac.cn (K.J.); 13563955956@163.com (M.Y.); t15095253866@163.com (Z.T.); 2College of Animal Science and Technology, Shandong Agricultural University, Tai’an 271018, China

**Keywords:** intramuscular fat, subcutaneous fat, fattening steer, multi-omics, integrating analysis

## Abstract

Intramuscular fat (IMF) represents a critical economic trait determining beef quality. Identifying biomarkers that distinguish intramuscular from subcutaneous fat facilitates our understanding of IMF deposition mechanisms and allows strategies to be developed to improve beef quality. Therefore, this study investigated the differences in gene and protein expressions as well as the lipid contents between these depots using integrated multi-omics analysis. We found that the intramuscular fat tissue exhibited enhanced expression of genes involved in calcium signaling and the glycolysis pathway at both the gene and protein levels, while exhibiting decreased expression at the protein level of genes involved in lipid metabolism. Eleven candidate proteins and three lipid species were established as IMF-specific candidates, which is valuable for developing breeding strategies or nutritional interventions to enhance IMF deposition in livestock.

## 1. Introduction

Intramuscular fat (IMF), commonly called marbling, is a key indicator of meat quality because the IMF content and composition directly affect tenderness, flavor intensity, sweet aroma, and palatability. In Wagyu, optimal beef flavor is associated with 35.4% IMF [1]. However, Hirai et al. [2] reported strong sweet and vegetable flavors in beef at approximately 40% IMF. In contrast to the IMF, subcutaneous fat (SCF) is generally considered a waste product of meat due to its low commercial value [3,4]. Consequently, selectively increasing the IMF content while reducing the SCF content is an effective strategy for improving meat quality.

Numerous previous studies have recognized that lipogenic ability is tissue-specific; IMF exhibits weakened adipogenic potential compared to SCF because of the smaller cell size, lower lipid content, and enzymatic activities than the SCF counterparts. Smith and Crouse [5] reported for the first time a smaller mean cell diameter (104 ± 2 μm) in isolated intramuscular adipocytes than in subcutaneous adipocytes (141 ± 5 μm) of Angus steers. Such disparity in cell size has been confirmed in cattle [6,7], pigs [8,9], and humans [10]. Siebert et al. [11] documented distinct fatty acid profiles in finished cattle, and IMF contained 44.8% saturated fatty acids (SFAs), 50.1% monounsaturated fatty acids (MUFAs), and 5.1% polyunsaturated fatty acids (PUFAs), compared with 43.1% SFAs, 54.9% MUFAs, and 2.0% PUFAs in SCF. Further in vivo and in vitro studies revealed that IMF contains less cytosolic lipid content [12,13,14] and diminishes the enzyme activities of leptin [14], adiponectin, and glycerol-3-phosphate dehydrogenase [15]. This reduced fat deposition efficiency in the IMF is attributed to its distinctive metabolic patterns and gene expression levels. In vitro, intramuscular adipocytes use more glucose than acetic acid to synthesize fatty acid carbon skeletons [5,16]. Several related genes, including *insulin-like growth factor II* (*IGF-II*) [14], *bone morphogenetic protein 4* (*BMP4*), and *pyruvate dehydrogenase kinase 4* [16], are upregulated in IMF. Contrastingly, downregulated genes enriched in the PPAR signaling pathway [17,18] include *fatty acid synthase* (*FASN*), *lipoprotein lipase* (*LPL*), and *peroxisome proliferator-activated receptor gamma* (*PPARG*).

Additionally, lipids other than fatty acids significantly affect fat deposition. Phosphatidylcholine (PC) supplementation inhibited the formation of hepatic lipid droplets and reduced *apolipoprotein E* (*APOE*) expression in mice fed a high-fat diet [19]. Conversely, adding phosphatidylethanolamine (PE) in intramuscular adipocytes promoted the accumulation of lipid droplets and upregulated the expression of *fatty acid-binding protein 4* (*FABP4*), *stearoyl-CoA desaturase* (*SCD*), and *PPARG* in chicken [20]. Recent results of untargeted lipidomics have revealed hundreds of different lipids correlated with IMF content in pigs [21,22,23] and donkeys [24]. However, the differences in lipid composition between bovine IMF and SCF remain poorly understood. Moreover, the correlation between lipid abundance and gene/protein expression remains unclear.

Hence, in this study, we aimed to elucidate the genetic mechanisms underlying IMF deposition by analyzing tissue-specific genes, proteins, and lipid metabolites of the IMF and SCF in cattle via integrated transcriptomics, proteomics, and lipidomics.

## 2. Materials and Methods

### 2.1. Animal Ethics and Management

All animal procedures met ARRIVE (Animal Research: Reporting of In Vivo Experiments) 2.0 guidelines and were approved by the Animal Ethics Committee of the Shandong Academy of Agricultural Sciences (Protocol No. SAAS-2024-067).

### 2.2. Animals, Management, and Sample Collection

Three thirty-month-old Angus steers (initial body weight of 573.6 ± 8.60 kg) were sourced from a commercial farm (36°24′ N, 116°23′ E) in Dezhou City of Shandong Province and were fed a basal diet with 5.41 MJ/kg of net energy gain during a 240-day fattening period. The basal diet was designed in compliance with NRC (2016) guidelines to fulfill the nutrient requirements of beef cattle. The composition and nutrient content of the basal diets are shown in Appendix A. Throughout the experimental period, all cattle were housed in the same natural environment and were fed twice daily at 07:00 and 17:00 with ad libitum access to potable water.

At the end of the experimental period, all fattened cattle were left unfed for 12 h, weighed (703.50 ± 11.45 kg), and slaughtered. Subcutaneous fat (SCF) from the back fat and IMF from the *longissimus thoracis et lumborum* muscles were separated. The fascia and connective tissue surrounding the adipose tissue were carefully removed. Subsequently, all samples were flash-frozen in liquid nitrogen and stored at −80 °C for subsequent analyses.

### 2.3. RNA Sequencing Analysis

Total RNA was extracted from 100 mg of tissue using the TRIzolTM reagent (Invitrogen, Carlsbad, CA, USA). The concentration and purity of the RNA samples were tested using NanoDrop 2000 (Thermo Fisher Scientific, Wilmington, DE, USA). Samples, with RNA integrity numbers (RINs) > 8.6 and 28S-to-18S rRNA ratios > 1.0, were subjected to RNA sequencing on the Illumina NovaSeq 6000 platform (Illumina, San Diego, CA, USA) at an approximately 6× sequencing depth. Six paired-end RNA libraries were obtained. Raw reads were filtered using fastp (v0.19.7) to remove adapters, low-quality bases (Q-score < 20), and poly N sequences. The clean reads were aligned to the Bos taurus reference genome (ARS-UCD2.0, NCBI Assembly. GCF_002263795.1) using HISAT2 (v2.2.1). Simultaneously, the mapping ratios to the genome were calculated, and the distribution of clean reads in exons and introns of the genome was analyzed. Gene expression was quantitatively analyzed using featureCounts (v1.5.0-p3), compared between IMF and SCF tissues, and expressed in terms of the fold change (FC); the corresponding logarithm (Log2Foldchange) was analyzed through pairwise comparisons using DESeq2 (v1.38.3) using R (version 4.2.2). The significance (*p*-values) was adjusted using the Benjamini–Hochberg approach. Differentially expressed genes (DEGs) were screened considering an adjusted *p*-value (*p*-adj) of 0.05 and an absolute Log2Foldchange of 1. Finally, Kyoto Encyclopedia of Genes and Genomes (KEGG) pathway enrichment was analyzed using ClusterProfiler (v4.6.2), and *p*-values were adjusted using the Benjamini–Hochberg method.

### 2.4. Tandem Mass Tag (TMT) Quantitative Protein Analysis

The frozen samples were individually ground in liquid nitrogen and lysed with PASP protein buffer, followed by 5 min of ultrasonication on ice. The supernatant was separated by centrifuging the sample at 12,000× *g* for 15 min at 4 °C and reduced using 10 mM DL-dithiothreitol and sufficient iodoacetamide. Then, the supernatant was completely mixed with pre-cooled acetone (1:4, *v*/*v*) and incubated for 2 h at −20 °C, followed by a centrifugation at 12,000× *g* for 15 min; next, the precipitate was collected and washed with 1 mL of cold acetone. Protein samples were dissolved in triethylammonium bicarbonate buffer supplemented with 8 M urea. The protein concentration was determined using a Bradford protein quantification kit (P0411M, Beyotime Biotechnology, Shanghai, China). Subsequently, samples were labeled by incubating them with TMT reagent dissolved in 41 μL of acetonitrile at room temperature for 2 h and freeze-dried to prepare a powder. The lyophilized powder was dissolved in solution A (2% acetonitrile and 98% water), and fractions were separated using a chromatographic column (BEH C18, Waters, Milford, MA, USA) at 45 °C in the L-3000 HPLC system. A 1 μg sample was injected into a C18 Nano-Trap column, and peptides were separated using an EASY-nLCTM 1200 UHPLC system (Thermo Fisher, Waltham, MA, USA) coupled with a Q ExactiveTM HF-X mass spectrometer (LC-MS/MS Analysis) with a 2.1 kV ion spray voltage, a transfer temperature of 320 °C, a scan range of *m*/*z* 350–1500, and a mass spectrometry resolution of 60,000 (200 *m*/*z*). The raw data of MS detection were analyzed using Proteome Discoverer 2.4 (PD2.4, Thermo Fisher), and qualified peptides with credibility ≥ 99% and FDR ≤ 1% were retained. The relative abundance of the peptides was calculated. The fold change and Log2Foldchange between IMF and SCF were calculated via t-tests, and differentially expressed proteins (DEPs) were screened based on |Log2Foldchange| > 0.46 and *p*-value < 0.05. GO, KEGG, and InterPro (IPR) analyses were conducted using Interproscan software (v5.50-84.0,) against non-redundant protein databases, including Clusters of Orthologous Groups (COGs), KEGG, and Protein Families (PFAMs).

### 2.5. Lipid Extraction and Lipidomic Analysis

Each sample (100 mg) was homogenized and transferred to a glass tube with a Teflon-lined cap. Pre-cooled methanol (0.75 mL) was added to it, and the mixture was vortexed. Pre-cooled methyl tert-butyl ether (MTBE, 2.5 mL) was added, followed by incubation at room temperature on a shaker for 1 h. Phase separation was induced by adding 0.625 mL of mass spectrometry-grade water, followed by incubation at room temperature for 10 min and centrifugation at 1000× *g* for 10 min. The organic phase was collected, and the lower phase was mixed with MTBE to further collect the organic phase. Combined organic phases were dried and dissolved in 100 μL of isopropanol for further analyses. Samples were injected onto a Thermo Accucore C30 column (150 × 2.1 mm, 2.6 μm) in a Vanquish UHPLC system (Thermo Fisher) coupled with an Orbitrap Q ExactiveTM HF mass spectrometer (Thermo Fisher) to perform ultra-high-performance liquid chromatography–tandem mass spectrometry (UHPLC-MS/MS); the column temperature was set at 40 °C. Buffer A contained 60% acetonitrile, 40% water, 10 mM ammonium acetate, and 0.1% formic acid, whereas buffer B contained 90% isopropanol, 10% acetonitrile, 10 mM ammonium acetate, and 0.1% formic acid. The solvent gradient was set as follows: 30% B, initial condition; 30% B, 2 min; 43% B, 5 min; 55% B, 5 min; 70% B, 11 min; 99% B, 16 min; and 30% B, 18 min. The Q ExactiveTM HF mass spectrometer was operated in positive or negative polarity modes. The raw data generated by UHPLC-MS/MS were processed using Compound Discoverer 3.01 (CD3.1, Thermo Fisher) for peak alignment, peak picking, and quantitation of each metabolite. Normalized data were used to predict molecular formulae based on additive ions, molecular ion peaks, and fragment ions. Lipids and their abundances were analyzed using the Lipidmaps and Lipidblast databases. In the comparative analysis of two tissues, the fold change (FC) of each lipid and the corresponding Log2Foldchange were calculated using the t-test. Lipids with VIP > 1, *p*-value < 0.05, and |Log2Foldchange| > 1 were considered differentially abundant lipids (DALs).

### 2.6. Bioinformatics Analysis

The Pearson correlations between DEPs and DALs were calculated using the cor.test function of the psych package (v. 2.5.6) in the R language. *p*-value < 0.05 and absolute r > 0.8 were considered to identify statistically significant correlations. Additionally, principal component analysis (PCA) and partial least squares discriminant analysis (PLS-DA) were performed using Stats (v. 1.0.12) and Ropls (v. 1.38.0). Volcano plots, heat maps, and Sankey plots were generated using the ggplot2 (v. 3.5.1), ggcorplot (v. 1.0.12), and ggsankey packages (v. 0.0.99), respectively.

## 3. Results

### 3.1. Transcriptomic Profiles and Differentially Expressed Genes (DEGs)

Transcriptome sequencing generated 40.52 GB of clean data (SCF: 20.46 GB; IMF: 20.06 GB; Table 1) and 270,107,766 clean reads, among which 136,396,756 were in the SCF group and 133,711,010 were in the IMF group. Q30 scores of all samples exceeded 90%, confirming data reliability. A total of 117,206,407 clean reads in SCF and 115,876,107 in IMF were mapping to the reference genome, respectively, with average mapping rates of 85.87% and 86.66% in the SCF and IMF groups. In addition, the mapping ratios of unique reads to the genome were 84.32% (SCF) and 84.97% (IMF).

In total, 15,573 genes were identified (Figure 1A), and 14,965 genes were expressed in both SCF and IMF tissue. A total of 311 and 297 genes were specifically expressed in IMF and SCF, respectively (Figure 1A). The PCA plot (Figure 1B) revealed clear differences in gene expression levels between the tissues. A comparative analysis of the IMF and SCF groups revealed 307 upregulated and 36 downregulated DEGs. KEGG enrichment of these 343 DEGs highlighted 18 significant pathways (Figure 1D), including adrenergic signaling in cardiomyocytes (*p*-adj = 6.1 × 10^−12^), calcium signaling (*p*-adj = 0.0000032), cGMP-PKG signaling (*p*-adj = 0.0000036), and cAMP signaling pathways (*p*-adj = 0.000065). Notably, DEGs associated with insulin secretion and glucagon synthesis were functionally linked to calcium, cGMP-PKG, and cAMP signaling (Figure 1E).

### 3.2. Proteomic Analysis and Differentially Expressed Proteins (DEPs)

TMT proteomics analysis identified 3909 proteins across the tissues (Appendix A). PCA revealed distinct proteomic profiles of IMF and SCF (Figure 2A). The results revealed 202 significantly more abundant and 79 less abundant DEPs in the IMF than in the SCF (Figure 2B). Subcellular localization classified these 281 DEPs as cytoplasmic (25.00%), nuclear (22.54%), cytoskeleton (11.07%), extracellular (10.25%), and cell membrane (7.38%) proteins, as well as thirteen other classes (Figure 2C). KEGG enrichment identified 14 significant pathways (Figure 2D); similar to the transcriptomic pathways, these pathways included adrenergic signaling in cardiomyocytes (*p* = 0.0000098), the calcium signaling pathway (*p* = 0.00052), and the cGMP-PKG signaling pathway (*p* = 0.0058). Additional DEP-enriched pathways included the glucagon signaling pathway (*p* = 0.0011) and glycolysis or gluconeogenesis (*p* = 0.00092).

### 3.3. Integrating Analysis of Transcriptome and Proteome

Furthermore, an integrated analysis was conducted to verify the reliability of the transcriptomic and proteomic data for the two fat tissues. Figure 3A showed that 3601 overlapping factors simultaneously appeared in the transcriptome and proteome, and both the mRNA and protein levels of 78 factors differed significantly between IMF and SCF (Figure 3B). A nine-quadrant diagram (Figure 3C) confirmed a strong correlation between the mRNA and protein. Among the 78 differentially expressed factors, 20 were significantly (*p*-adjusted < 0.05) enriched in adrenergic signaling in cardiomyocytes, cardiac muscle contraction, the calcium signaling pathway, the cGMP-PKG signaling pathway, the cAMP signaling pathway, and the oxytocin signaling pathway (Figure 3D). Furthermore, myosin light chain 3 (MYL3), MYLK (myosin light chain kinase), myosin heavy chain 7 (MYH7), troponin C1, slow skeletal and cardiac type (TNNC1) tropomyosin 2 (TPM2), ATPase sarcoplasmic/endoplasmic reticulum Ca^2+^ transporting 2 (ATP2A2), and ryanodine receptor 1 (RYR1) were related to the calcium signaling pathway. Enolase 3 (ENO3), fructose-bisphosphatase 2 (FBP2), and glycogen phosphorylase (PYGM) were involved in the glucagon signaling pathway and glycolysis. Interestingly, both their mRNA and protein levels were higher in the IMF than in the SCF (Figure 3E). Contrastingly, three lipid metabolism-related factors were suppressed in IMF, which included fatty acid-binding protein 4 (FABP4), stearoyl-CoA desaturase (SCD), and apolipoprotein E (APOE).

### 3.4. Untargeted Lipidomics and Differential Abundance Lipids (DALs)

UPLC-MS analysis revealed 360 and 150 lipids in the negative and positive polarity modes, respectively (Figure 4A and Appendix A), which include 19 acylcarnitine (Acars), 20 ceramides (Cers), 23 cardiolipins (CLs), 9 diacylglycerols (DAGs), 19 free fatty acids (FAs), and 3 monosialodihexosylgangliosides (GMs), as well as 18 phosphatidic acids (PAs), 1 phosphatidylethanol, 113 phosphatidylcholines (PCs), 140 phosphatidylethanolamines (PEs), 18 phosphatidylglycerols (PGs), 33 phosphatidylinositols (PIs), 15 phosphatidylserines (PSs), 41 sphingomyelines (SMs), and 12 triacylglycerols (TAGs). The abundance of eighty-eight lipids was distinctive (Figure 4B) between the IMF and SCF. In the negative polarity mode, 37 DALs were significantly decreased in IMF compared with SCF counterparts, which included two FAs, one PA, ten PCs, nine PEs, two PGs, six PIs, five PSs, and two SMs, whereas 16 DALs were significantly increased in the IMF (Figure 4C,D). In the positive polarity mode, 31 of 35 lipids were significantly increased in the IMF, and only 4 lipids were decreased (Figure 4E,F). No KEGG pathway was significantly enriched in the DALs.

### 3.5. Pearson Correlation Analysis

The top 20 lipids sorted by Log2foldchange values in both negative and positive polarity mode were selected for Pearson correlation analysis. Significant correlations among Acar 18:2, Acar 20:4, Acar 18:3, Hydroxyoetadeeenoylearnitine, Acar 16:1, and PI (18:0/20:1) were detected (Figure 5A). PS (18:1/20:3) was negatively correlated with PI (18:0/18:1) and Acar 20:4.

The Pearson correlation (Figure 5B) reflected significant positive correlations among the calcium signaling pathway and glycolysis-associated factors, including ATP2A2, MYL3, MYH7, MYLK, TNNC1, TPM2, ENO3, FBP2, and PYGM, while there was a negative correlation between these ten factors and FABP4, SCD, and APOE. Furthermore, APOE was positively correlated with PI (18:1/20:4) and PC (15:0/18:2), whereas FBP2 was positively correlated with LPE18:0. PI (18:1/20:4) was negatively associated with ENO3 and PYGM, whereas PC (15:0/18_2) was negatively associated with TPM and MYLK.

## 4. Discussion

Recent reports revealed significant differences between the IMF and SCF in livestock, which reflected variable cell morphology, hormone levels, metabolic capacity, fatty acid composition, and gene expression. Identifying potential IMF-related biomarkers was crucial for developing targeted strategies for enhancing the IMF content in meat. The present study revealed that the DEGs and DEPs were enriched in calcium signaling and glycolysis pathways, and MYH7, MYL3, MYLK, TNNC1, TPM2, ATP2A, ENO3, FBP2, and PYGM were upregulated in IMF. However, FABP4, APOE, and SCD were downregulated. In addition, the abundances of LPE18:0 increased in IMF, yet PI (18:1/20:4) and PC (15:0/18:2) decreased.

### 4.1. The Tissue-Specific Factors Enriched in the Calcium Pathway Between IMF and SCF in Steers

In the skeletal muscles, intramuscular adipocytes primarily originate from mesenchymal stem cells (MSCs) in the mesoderm. MSCs, highly expressing *platelet-derived growth factor receptor alpha* (*PDGFRα*^+^), differentiated into adipocytes [25,26], while those expressing *myogenic factor 5* (*MYF5*^+^) differentiated into muscle cells [27]. IMF tissue shared the same origin as myocytes and exhibited higher expression levels of genes enriched in calcium signaling pathways, such as *MYH7*, *MYL3*, *TNNC1*, and *TPM2*. Similarly, actin cytoskeleton components were associated with intramuscular fat in livestock [28,29,30]. *TPM2* expression was upregulated in the intramuscular adipose tissue of Hanwoo bulls compared to the omental adipose tissue [31]. MYL3 [32], MYH7 [33,34], and TNNC1 [35] affected fat formation or meat quality. Feng et al. [36] predicted *MYH7* and *TNNC1* as hub genes affecting IMF deposition.

### 4.2. The Tissue-Specific Factors Related to Glycolysis and Lipid Metabolism in IMF and SCF of Steers

The glycolytic ability differed between IMF and SCF, and ENO3, FBP2, and PYGM were upregulated in IMF compared to SCF, at both mRNA and protein levels. Enhanced glycolytic ability was detected in the high-IMF muscle tissues of Dingyuan pigs [37] and Guangling donkeys [38]. ENO3 [39], FBP2 [40], and PYGM [41,42] were positively correlated with muscle type, intramuscular fat, or bovine tenderness. Furthermore, Ma et al. [43] reported that *PGAM2* overexpression in goat intramuscular adipocytes upregulated *PPARG*, *CCAAT/Enhancer-binding protein α (C/EBPα*), and *lipoprotein lipase* (*LPL*) and promoted lipid accumulation. Additionally, two subunits of phosphorylase kinase (PHK), PHKA1 and PHKB, were highly expressed in IMF with a slight increase in *PHKG1* mRNA. PHK activated glycogen breakdown through a Ca^2+^-dependent reaction. Notably, inhibition of *PHKG1* expression enhances preadipocyte adipogenesis via upregulation of *PPARG*, *CEBPβ*, and *FABP4* [44].

In contrast, in IMF, there was a decrease in the expression of genes related to lipid metabolism in IMF at both the gene and protein levels, such as FABP4, APOE, and SCD. Both FABP4 and SCD are considered the main IMF-associated loci in Qinchuan cattle [45] and Duroc pigs [46]. Although no direct evidence links APOE to the IMF, Song et al. [47] reported that silencing APOE inhibited *PPARG*, *FABP4*, and *CD36* expression, leading to a reduction in the lipid accumulation in a cell co-culture model comprising muscle cells and intramuscular adipocytes. The inverse correlation between glycolysis and lipid metabolism in the IMF may reflect glucose diversion toward de novo lipogenesis.

### 4.3. The Tissue-Specific Lipids in IMF and SCF of Steers

Intramuscular and subcutaneous fat exhibited significantly variable lipid composition. Bosch et al. [48] reported that IMF contained higher MUFA (e.g., oleic acid) and lower PUFA contents (e.g., linoleic acid) than SCF. Similarly, the oleic acid (C18:1) content was higher in pigs [22,48] with high IMF contents. Conversely, pentadecanoic acid, heptadecanoic acid, linoleic acid, α-linolenic acid, cis-11,14-eicosadienoic acid, and arachidonic acid were lower in the high-IMF group. Recent studies have found that the phospholipid metabolism-related genes and lipids regulate IMF deposition [9,20,37,49]. Phosphatidylcholine negatively regulated IMF deposition, whereas phosphatidylethanolamine and phosphatidylserine promoted it [20]. In this study, 65 phospholipids among 88 lipids differed between the two tissues, which included 22 PCs, 19 PEs, 11 PIs, 6 PSs, 5 SMs, and 2 PGs. Meng and Yan [22] identified 112 lipids, containing 49 phospholipids related to IMF, in Chinese Erhualian pigs. Hou et al. [21] identified 57 lipids in Laiwu pigs and suggested PE (18:1e/20:4) and PC (16:1e/20:4) to be potential IMF-related biomarkers. In our study, PC (15:0/18:2) and PI (18:1/20:4) were reduced in IMF compared to SCF, while LPE 18:0 was enhanced. Furthermore, PC (15:0/18:2) was negatively correlated with beef marbling [50]. PI (18:1/20:4) was reported to be negatively correlated with IMF deposition in Jingyuan chickens [51]. Conversely, LPE 18:0 [52] could promote lipolysis via cAMP-dependent phosphorylation of hormone-sensitive lipase.

### 4.4. Limitation

An apparent limitation of this study was the relatively small sample size (*n* = 3 steers), which may restrict the statistical power in identifying some genes or proteins with moderate expression levels. Future investigations with larger cohorts or cellular-level gene silencing and overexpression would be required to further validate their roles in intramuscular fat deposition.

## 5. Conclusions

In conclusion, this study demonstrated that IMF tissue promoted calcium signaling pathway and glycolytic potential compared with SCF, with attenuated lipid metabolism. ATP2A2 (SERCA2), ENO3, FBP2, MYH7, MYL3, MYLK, PYGM, TNNC1, TPM2, and LPE (18:0) were associated with the suppression of fat accumulation. Conversely, FABP4, SCD, APOE, PI (18:1/20:4), and PC (15:0/18:2) were positively correlated with fat accumulation. These key genes and proteins affecting intramuscular fat content could be incorporated into genomic selection strategies. Estimating allele frequencies and breeding values for these candidates would facilitate the selection of animals with a genetic predisposition for higher IMF. Furthermore, dietary interventions—such as supplementation with phosphatidylethanolamine or calcium—could increase intramuscular fat contents. These findings provide valuable insights for developing strategies to enhance IMF content in livestock.

## Figures and Tables

**Figure 1 animals-15-02733-f001:**
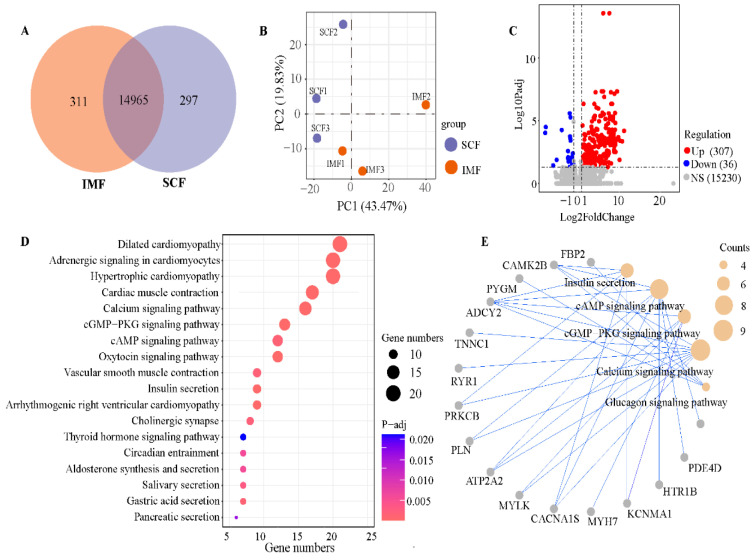
The differential gene expression analysis between intramuscular and subcutaneous fat tissue. (**A**) The Venn diagram showed 14,965 genes co-expressed in both intramuscular fat (IMF) and subcutaneous fat (SCF) tissues, with 311 IMF-specific genes and 297 SCF-specific genes. (**B**) Principal component analysis showed different gene expression patterns between IMF and SCF. (**C**) The volcano plot identified 307 upregulated differentially expressed genes (DEGs) and 36 downregulated DEGs in IMF compared with SCF. (**D**) Kyoto Encyclopedia of Genes and Genomes (KEGG) pathway analysis showed that 343 DEGs were enriched in 18 significant KEGG pathways. (**E**) There was a significant enrichment of DEGs in 5 metabolism-related KEGG pathways.

**Figure 2 animals-15-02733-f002:**
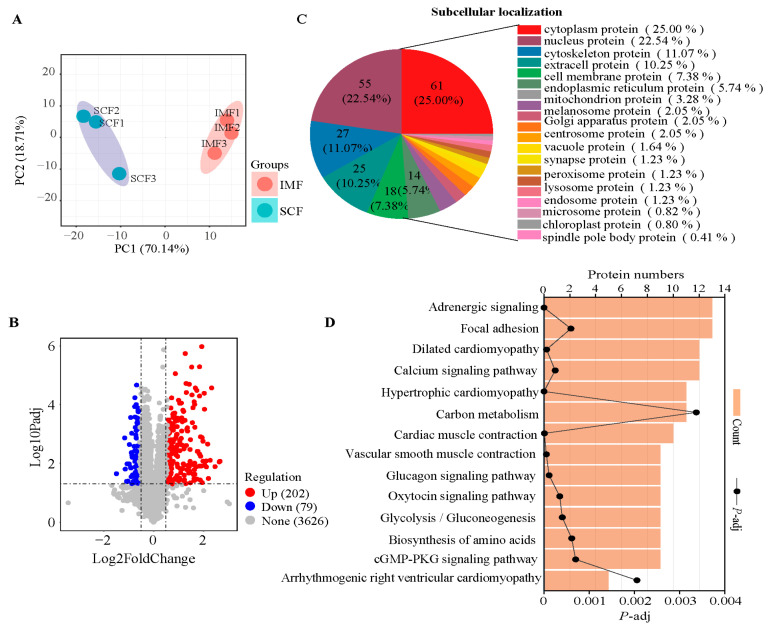
The differential protein expression between subcutaneous and intramuscular fat tissues. (**A**) Principal component analysis showed significant differences in protein expression between intramuscular fat (IMF) and subcutaneous intramuscular fat (SCF) tissues. (**B**) The volcano plot showed that 202 differentially expressed proteins (DEPs) were upregulated and 79 DEPs were downregulated in IMF compared with SCF. (**C**) The subcellular localization of the 281 DEPs showed that proteins were mainly divided into 18 categories, including cytoplasmic proteins, nuclear proteins, and cytoskeletal proteins. (**D**) Functional enrichment analysis confirmed that 14 pathways were significantly enriched by 281 DEPs.

**Figure 3 animals-15-02733-f003:**
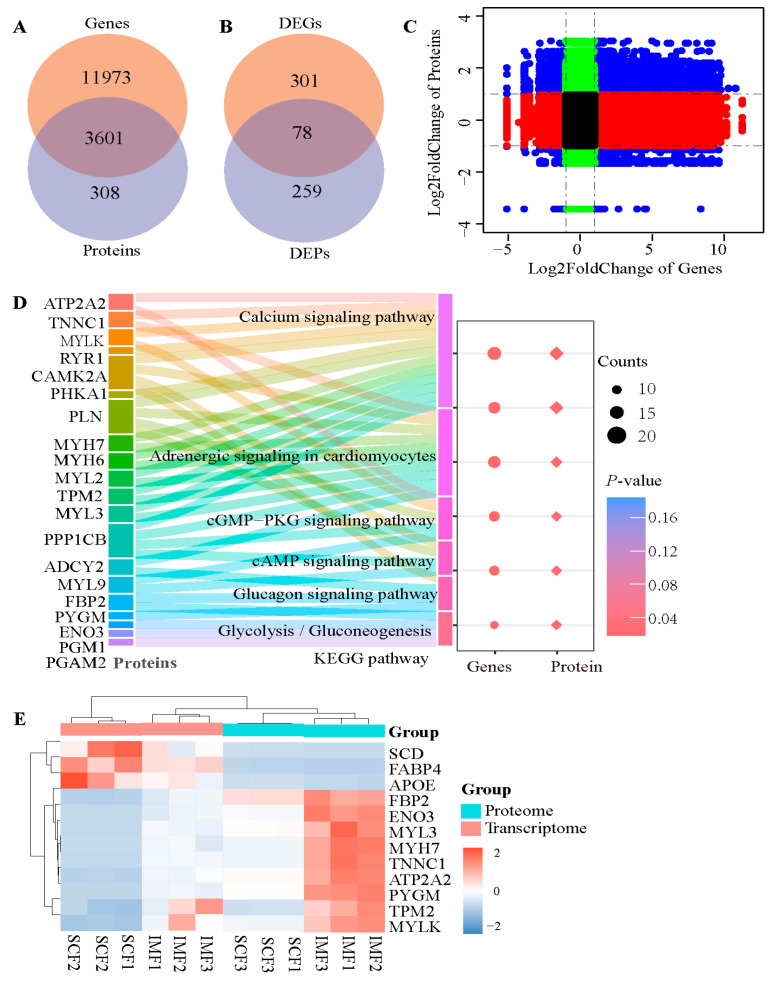
Integrated analyses of transcriptome and proteome from intramuscular and subcutaneous fat tissues in fattening cattle. (**A**) The Venn diagram showed that 3601 out of 3909 proteins in the proteome have corresponding coding genes in two fat tissues. (**B**) Both the gene and protein expression levels of 78 factors were significantly different between intramuscular fat (IMF) and subcutaneous fat (SCF) tissues. (**C**) The quadrant diagram of the correlation between 3601 genes and their coding proteins. (**D**) Six significant KEGG pathways were enriched by 20 overlapping factors in both transcriptome and proteome. (**E**) The cluster heat map showed that the gene and protein expression levels of nine candidate factors increased, while those of three candidates decreased in IMF compared with SCF. APOE, apolipoprotein E; ATP2A2, sarcoplasmic/endoplasmic reticulum Ca^2+^ transporting 2; ENO3, enolase 3; FABP4, fatty acid-binding protein 4; FBP2, fructose-bisphosphatase 2; MYH7, myosin heavy chain 7; MYL3, myosin light chain 3; MYLK, myosin light chain kinase; PYGM, glycogen phosphorylase; SCD, stearoyl-CoA desaturase; TNNC1, troponin C1; TPM2, tropomyosin 2.

**Figure 4 animals-15-02733-f004:**
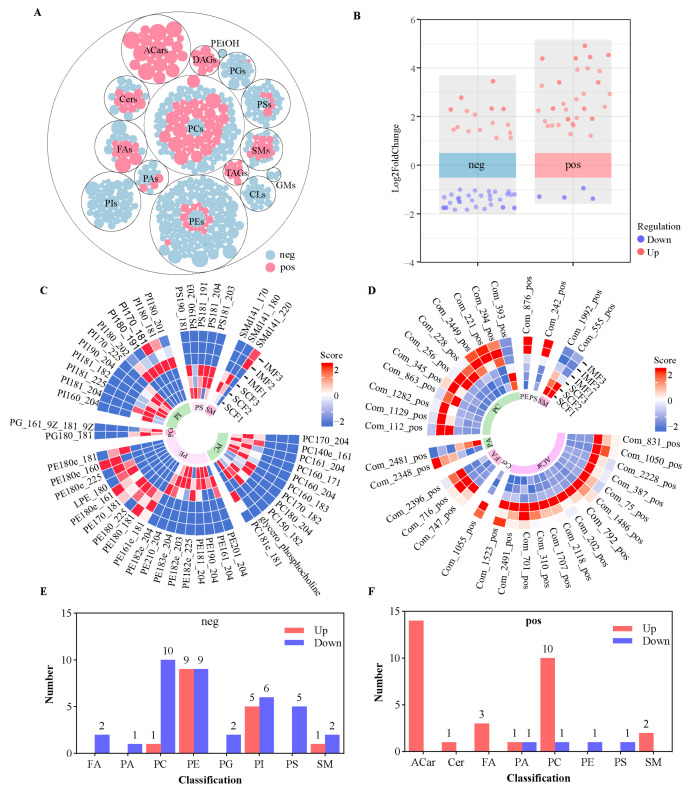
Differential lipid metabolites between subcutaneous and intramuscular fat tissues of fattening cattle. (**A**) A total of 510 lipids were identified in subcutaneous and intramuscular fat tissues, containing 340 positive and 170 negative ions, and were divided into 15 subclasses. (**B**) Differential abundance analysis indicated 53 differentially abundant lipids (DALs) in negative-ion mode (neg) and 35 DALs in positive-ion mode (pos) between IMF and SCF. (**C**) The circular heat map showed the abundance of 53 DALs in neg mode. The abundance of 37 DALs increased in the IMF, yet 16 DALs decreased. (**D**) A total of 35 DALs in pos mode were identified between IMF and SCF. Among them, the abundance of 31 DALs increased in the IMF. (**E**) A classified bar chart of 53 DALs in a negative polarity model. (**F**) A classified bar chart of 35 DALs in a positive polarity mode. Acars, acylcarnitine; Cers, ceramides; CLs, cardiolipins; DAGs, diacylglycerols; FAs, free fatty acids; GM3, monosialodihexosylganglioside; PAs, phosphatidic acids; PetOH, phosphatidylethanol; PCs, phosphatidylcholines; PE, phosphatidylethanolamine; PG, phosphatidylglycerol; PI, phosphatidylinositol; PS, phosphatidylserine; SM, sphingomyelin; TAGs, triacylglycerols; neg: negative polarity mode; pos: positive polarity mode.

**Figure 5 animals-15-02733-f005:**
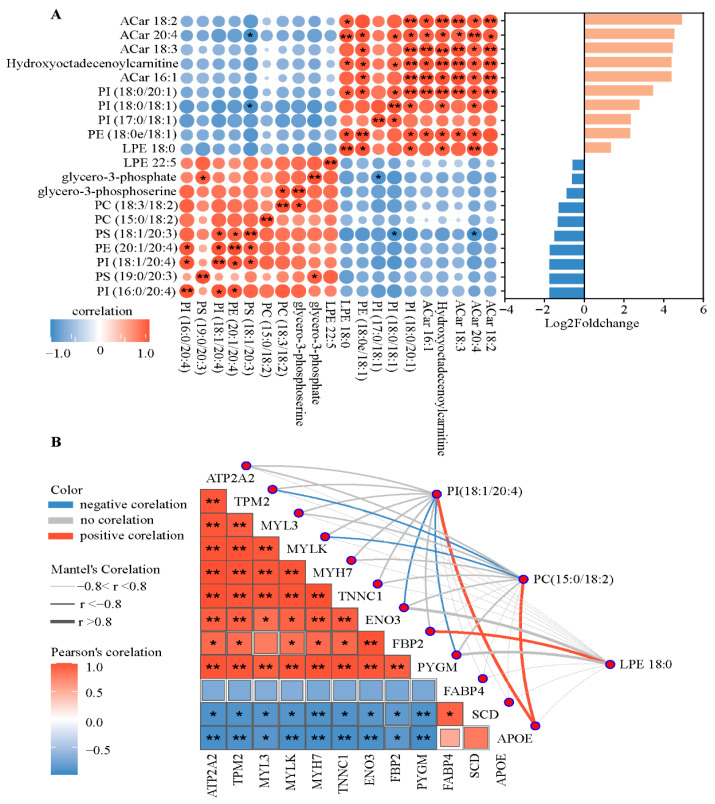
Pearson’s correlation analysis of differentially abundant proteins and lipids related to IMF in finishing steers. (**A**) The top 20 differentially expressed lipids (DALs) ranked in absolute log2foldchange were selected and subjected to correlation analysis. (**B**) A heat map of the correlations between 12 candidate differentially abundant proteins and 3 candidate differentially abundant lipids. ** indicates an absolute correlation coefficient > 0.8, and *p*-value < 0.01. * indicates an absolute correlation coefficient > 0.8, and *p*-value < 0.05. Acar, acylcarnitine; APOE, apolipoprotein E; ATP2A2, sarcoplasmic/endoplasmic reticulum Ca^2+^ transporting 2; ENO3, enolase 3; FABP4, fatty acid-binding protein 4; FBP2, fructose-bisphosphatase 2; MYH7, myosin heavy chain 7; MYL3, myosin light chain 3; MYLK, myosin light chain kinase; PCs, phosphatidylcholines; PE, phosphatidylethanolamine; PI, phosphatidylinositol; PS, phosphatidylserine; PYGM, glycogen phosphorylase; SCD, stearoyl-CoA desaturase; TNNC1, troponin C1; TPM2, tropomyosin 2.

**Table 1 animals-15-02733-t001:** Statistics of total and mapping reads of subcutaneous and intramuscular fat tissues in fattening cattle, which were mapped to the reference genome.

Groups	Samples	Raw Bases	Clean Bases	Clean Reads	Mapping Reads to Genome	Mapping Ratio to Genome	Unique Reads	Unique Mapping Ratio	Error Rate	Q30
SCF	SCF1	7.24 G	7.05 G	47,018,464	41,812,006	88.93%	41,058,526	87.32%	0.03	91.29%
	SCF2	6.92 G	6.69 G	44,590,924	35,766,667	80.21%	35,101,512	78.72%	0.03	91.48%
	SCF3	7.03 G	6.72 G	44,787,368	39,627,734	88.48%	38,924,720	86.91%	0.03	91.70%
IMF	IMF1	6.89 G	6.75 G	44,984,872	39,267,491	87.29%	38,519,307	85.63%	0.03	91.37%
	IMF2	6.84 G	6.65 G	44,301,772	37,477,298	84.60%	36,686,678	82.81%	0.03	91.96%
	IMF3	6.78 G	6.66 G	44,424,366	39,131,318	88.09%	38,414,966	86.47%	0.03	91.74%

## Data Availability

Transcriptomic sequence reads for all samples are available under the National Center for Biotechnology Information with accession project number PRJNA1308542 (https://www.ncbi.nlm.nih.gov/bioproject/PRJNA1308542) accessed on 20 August 2025. Proteomic and lipidomic data can be found in the Appendix A.

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
