# Peer review of "Comparative Analyses of Gene and Protein Expressions and the Lipid Contents in Intramuscular and Subcutaneous Fat Tissues in Fattening Steers"

_animals, 2025, doi:10.3390/ani15182733_

Round 1
Reviewer 1 Report
Comments and Suggestions for Authors
I reviewed the article entitled „Comparative analyses of gene and protein expression, and the lipid contents in intramuscular and subcutaneous fat tissues in fattening steers”.
The aim of the study was to investigate the differences in gene and protein expression, and the lipid contents between intramuscular and subcutaneous fat depots, using multi-omics analysis.
Row 17: after „glycolysis pathway” the sentenxe seems somehow unfinished. Moreover, the next sentence starts with „And...”, fact which is improperly.
Row 32-33: PI and PC were ristly used without explaining what they mean.
Row 41: „muti-omics”?
Row 47: instead ; use full stop
Row 55: delete comma after „for the first time”
Row 56: delete the extraspace after 141
Row 61: in vivo, in vitro - use Italic
Row 65: in vitro - use Italic
A scheme for the biochemical mechanisms presented above is wellcome.
Row 80: Full stop after „poorly understood”
Row 301: delete comma after IMF and put full stop. Start a new sentence with ”However”
A general recommendation for all chapter 4 is to use schemes and tables in order to make the information easily to be red and understood.
Author Response
Responses to the Reviewers
Reviewer1:
Comments and Suggestions for Authors
I reviewed the article entitled „Comparative analyses of gene and protein expression, and the lipid contents in intramuscular and subcutaneous fat tissues in fattening steers”.
The aim of the study was to investigate the differences in gene and protein expression, and the lipid contents between intramuscular and subcutaneous fat depots, using multi-omics analysis.
Response:
Thank you for your valuable suggestions for improving the paper.
Row 17: after „glycolysis pathway” the sentenxe seems somehow unfinished. Moreover, the next sentence starts with „And...”, fact which is improperly.
Response:
We revised the content in row 17:
“We found that the intramuscular fat tissue exhibited enhanced both gene and protein expression levels involved in calcium signaling and glycolysis pathway, while exhibiting decreased protein expression levels involved in lipid metabolism.”
Row 32-33: PI and PC were ristly used without explaining what they mean.
Response:
We revised the content in row 32-33:
“The abundances of both phosphatidylinositol (PI) (18:1/20:4) and phosphatidylcholine (PC) (15:0/18:2) were positively correlated with APOE.”
Row 41: „muti-omics”?
Response:
We revised the spelling mistakes in row 41.
Row 47: instead ; use full stop
Response:
We revised the content in row 47.
Row 55: delete comma after „for the first time”
Response:
We revised the contents in row 55.
Row 56: delete the extraspace after 141
Response:
We revised the contents in row 56.
Row 61: in vivo, in vitro - use Italic
Response:
We revised the font format of in vivo to Italic in row 61.
Row 65: in vitro - use Italic
Response:
We revised the contents in row 65.
A scheme for the biochemical mechanisms presented above is wellcome.
Response:
The detailed biochemical mechanism can be found in reference [14]. As described by Wang et al. (2013), both subcutaneous adipocytes (SCA) and intramuscular adipocytes (IMA) are committed from mesenchymal stem cells. The differentiation program of SCA is faster and stimulated mainly by bone morphogenetic protein 2 (BMP2), followed by PPARγ, C/EBPα and other genes related to adipogenesis, and the accumulation of lipids in subcutaneous adipocytes are mainly dependent on the transport of exogenous fatty acids (FA). In IMA, however, the differentiation program is slower and stimulated mainly by BMP2 and BMP4. In contrast to SCA, IMA mainly use glucose for lipogenesis, and the accumulation of cellular lipids is less than that of SCA in the late phase of differentiation.
Figure. Schematic representation of the difference between i.m. and s.c. preadipocytes on differentiation patterns. (referred from Wang et al. 2013)
Row 80: Full stop after „poorly understood”
Response:
We revised the contents in row 80.
Row 301: delete comma after IMF and put full stop. Start a new sentence with ”However”
Response:
We revised the contents in row 328.
A general recommendation for all chapter 4 is to use schemes and tables in order to make the information easily to be red and understood.
Response:
We thank the reviewer for this suggestion.
We fully agree that visual elements can significantly enhance the clarity of the findings. Upon careful consideration, we found that the content in chapter 4 primarily explaining the complex interactions and biological functions of the pathways, genes, and lipids we identified. With the guidance of a professional English editor, we carefully revised all the content and optimized the text structure to enhance readability.
Reviewer 2 Report
Comments and Suggestions for Authors
Dear Authors,
Your study addresses an important question in animal science and meat quality and that is the molecular differences between intramuscular fat and subcutaneous fat . The use of integrated multi-omics (transcriptomics, proteomics, lipids) is a major strength and provides a comprehensive view of the metabolic distinctions between these two tissues. The identification of calcium signaling and glycolysis pathways as key factors is particularly interesting, and the manuscript is well-supported by detailed methodology and a thorough literature review.
There are several areas where the manuscript would benefit from revision. The most important limitation is the small sample size (n=3 steers), which restricts the statistical robustness of your findings. Although analyses are resource-intensive, conclusions should be framed more cautiously to reflect this constraint. In particular, while you identify candidate genes, proteins, and lipids, these should not yet be described as confirmed “biomarkers.” Instead, please present them as potential candidates that warrant further validation in larger cohorts or with functional studies (qPCR, western blotting, gene silencing/overexpression).
The writing and structure of the manuscript could also be improved. Some sections are repetitive, and phrasing is occasionally awkward, which affects clarity. Streamlining the results and discussion will make the paper more accessible to readers. Figures, while comprehensive, are very dense; captions should be expanded so that readers unfamiliar with omics data can easily interpret them.
I encourage you to strengthen the discussion of practical implications. For example, how might the identified pathways or candidate molecules be used in breeding strategies or nutritional interventions to enhance IMF deposition?
Your study is scientifically interesting and presents valuable data. With revisions to improve clarity, tone down overstatements, and emphasize limitations, the manuscript could make a strong contribution to understanding IMF biology.
Comments on the Quality of English LanguageThe quality of English language in this manuscript is adequate, but needs significant improvement in clarity and readability. While the technical terminology is appropriate, the text often suffers from grammatical errors, awkward phrasing, and overly long or repetitive sentences. Repetition is found in the abstract, results, and discussion, which unnecessarily lengthens the manuscript. Some figure captions are too short to guide the reader through complex data. Overall, the content is understandable, but needs revision to improve flow, eliminate redundancies, and ensure that the language meets international publication standards. Professional English editing is strongly recommended to refine grammar, sentence structure, and scientific style.
Author Response
Reviewer2:
Comments and Suggestions for Authors
Dear Authors,
Your study addresses an important question in animal science and meat quality and that is the molecular differences between intramuscular fat and subcutaneous fat . The use of integrated multi-omics (transcriptomics, proteomics, lipids) is a major strength and provides a comprehensive view of the metabolic distinctions between these two tissues. The identification of calcium signaling and glycolysis pathways as key factors is particularly interesting, and the manuscript is well-supported by detailed methodology and a thorough literature review.
Response:
We sincerely thank the reviewer for their positive comments and valuable suggestions.
There are several areas where the manuscript would benefit from revision. The most important limitation is the small sample size (n=3 steers), which restricts the statistical robustness of your findings. Although analyses are resource-intensive, conclusions should be framed more cautiously to reflect this constraint. In particular, while you identify candidate genes, proteins, and lipids, these should not yet be described as confirmed “biomarkers.” Instead, please present them as potential candidates that warrant further validation in larger cohorts or with functional studies (qPCR, western blotting, gene silencing/overexpression).
Response:
We wholeheartedly thank the reviewer for this suggestion regarding the sample size, and we agree that it represents a key limitation of the present study. We have modified the inappropriate expression “biomarker” to “potential candidates” or “candidate molecules”. Validation in larger cohorts or with functional studies (qPCR, Western blotting, gene silencing/overexpression) will be performed in a further study.
The writing and structure of the manuscript could also be improved. Some sections are repetitive, and phrasing is occasionally awkward, which affects clarity. Streamlining the results and discussion will make the paper more accessible to readers. Figures, while comprehensive, are very dense; captions should be expanded so that readers unfamiliar with omics data can easily interpret them.
Response:
We sincerely thank you for this insightful comment.
A professional English editor from MDPI Author Services make extensive revisions to our article, correcting grammar errors and improving sentence structure and streamlining the results and discussion.
We have expanded all the captions to make them easy to understand for all readers.
In lines 205-216:“Figure 1. The differential gene expression analysis between intramuscular and subcutaneous fat tissue. A: The venn diagram showed 14 965 genes co-expressed in both intramuscular fat (IMF) and subcutaneous fat (SCF) tissues, with 311 IMF-specific genes and 297 SCF-specific genes. B. Principal component analysis showed different gene expression patterns between IMF and SCF. C. The volcano plot identified 307 up-regulated differentially expressed genes (DEGs) and 36 down-regulated DEGs in IMF compared with SCF. D. Kyoto Encyclopedia of Genes and Genomes (KEGG) pathway analysis showed that 343 DEGs were enriched in 18 significant KEGG pathways. E. There was a significant enrichment of DEGs in 5 metabolism-related KEGG pathways.”
In lines 226-233: “Figure 2. The differential protein expression between subcutaneous and intramuscular fat tissues. A: Principal component analysis showed significant differences in protein expression between in-tramuscular fat (IMF) and subcutaneous intramuscular fat (SCF) tissues. B. The volcano plot showed that 202 differentially expressed proteins (DEPs) were upregulated and 79 DEPs were downregulated in IMF compared with SCF. C. The subcellular localisation of the 281 DEPs shown that proteins were mainly divided into 18 categories, including cytoplasmic proteins, nuclear pro-teins, and cytoskeletal proteins. D. Functional enrichment analysis confirmed that 14 pathways significantly enriched by 281 DEPs.”
In lines254-265: “Figure 3. Integrated analyses of the transcriptome and proteome from intramuscular and subcu-taneous fat tissues in fattening cattle. A. The Venn diagram showed that 3601 out of 3909 proteins in the proteome have corresponding coding genes in two fat tissues. B. Both the gene and protein expression levels of 78 factors were significantly different between intramuscular fat (IMF) and subcutaneous fat (SCF) tissues. C. The quadrant diagram of the correlation between 3601 genes and their coding proteins. D. Six significant KEGG pathways were enriched by 20 overlapping in both the transcriptome and proteome. E. The cluster heat map showed increased gene and protein expression levels of 9 candidate factors, with a decrease in that of three candidates in IMF com-pared with SCF. APOE, apolipoprotein E; ATP2A2, sarcoplasmic/endoplasmic reticulum Ca2+ transporting 2; ENO3, enolase 3; FABP4, fatty acid-binding protein 4; FBP2, fructose-bisphosphatase 2; MYH7, myosin heavy chain 7; MYL3, myosin light chain 3; MYLK, myosin light chain kinase; PYGM, glycogen phosphorylase; SCD, stearoyl-CoA desaturase. TNNC1, troponin C1; TPM2, tropomyosin 2.”
In lines 282-295:“Figure 4. Differential lipid metabolites between subcutaneous and intramuscular fat tissues of fat-tening cattle. A: A total of 510 lipids were identified in subcutaneous and intramuscular fat tissues, containing 340 positive and 170 negative ions, and were divided into 15 subclasses. B: Abundance differences analysis indicated 53 differentially abundant lipids (DALs) in negative ion mode (neg) and 35 DALs in positive ion mode (pos) between IMF and SCF. C: The circular heatmap showed the abundance of 53 DALs in neg mode. The abundance of 37 DALs increased in IMF, yet that 16 DALs decreased. D: A total of 35 DALs in pos mode were identified between IMF and SCF. Among them, the abundance of 31 DALs increased in IMF. E: A classified bar chart of 53 DALs in negative polarity model. F.: A classified bar chart of 35 DALs in positive polarity mode. Acars, acylcarnitine; Cers, ceramides; CLs, cardiolipins; DAGs, diacylglycerols; FAs, free fatty acids; GM3, monosialodihexosylganglioside. PAs, phosphatidic acid; PetOH, phosphatidylethanol; PCs, phos-phatidylcholine; PE, phosphatidylethanolamine; PG, phosphatidylglycerol; PI, phosphatidylinosi-tol; PS, phosphatidylserine; SM, sphingomyelin; TAG, triacylglycerols. Neg: negative polarity mode; Pos: positive polarity mode.”
In lines 310-320:“Figure 5. The Pearson’s correlation analysis of differentially abundant proteins and lipids related to IMF in finishing steers. A: The top 20 differentially expressed lipids (DALs) ranked in absolute log2foldchange were selected, and were subjected to correlation analysis. B: A heat map of the correlations between 12 candidate differentially abundant proteins and 3 candidate differentially abundant lipids. ** indicates an absolute correlation coefficient > 0.8, and P-value < 0.01. * indicates an absolute correlation coefficient > 0.8, and P-value < 0.05. Acar, acylcarnitine; APOE, apolipoprotein E; ATP2A2, sarcoplasmic/endoplasmic reticulum Ca2+ transporting 2; ENO3, enolase 3; FABP4, fatty acid-binding protein 4; FBP2, fruc-tose-bisphosphatase 2; MYH7, myosin heavy chain 7; MYL3, myosin light chain 3; MYLK, myosin light chain kinase; PCs, phosphati-dylcholine; PE, phosphatidylethanolamine; PI, phosphatidylinositol; PS, phosphatidylserine. PYGM, glycogen phosphorylase; SCD, stearoyl-CoA desaturase. TNNC1, troponin C1; TPM2, tropomyosin 2.”
I encourage you to strengthen the discussion of practical implications. For example, how might the identified pathways or candidate molecules be used in breeding strategies or nutritional interventions to enhance IMF deposition?
Response:
We thank the reviewer for this valuable suggestion.
We strengthen the discussion of practical implications in lines 396-402: “These key genes and proteins affecting intramuscular fat content could be incorpo-rated into genomic selection strategies. Estimating allele frequencies and breeding values for these candidates would facilitate the selection of animals with a genetic predisposition for higher IMF. Furthermore, dietary interventions—such as supple-mentation with phosphatidylethanolamine or calcium—could increase intramuscular fat contents. These findings provide valuable insights for developing strategies to enhance IMF content in livestock.”
Your study is scientifically interesting and presents valuable data. With revisions to improve clarity, tone down overstatements, and emphasize limitations, the manuscript could make a strong contribution to understanding IMF biology.
Response:
We sincerely thank the reviewer for their positive comments and valuable suggestions.
Comments on the Quality of English Language
The quality of English language in this manuscript is adequate, but needs significant improvement in clarity and readability. While the technical terminology is appropriate, the text often suffers from grammatical errors, awkward phrasing, and overly long or repetitive sentences. Repetition is found in the abstract, results, and discussion, which unnecessarily lengthens the manuscript. Some figure captions are too short to guide the reader through complex data. Overall, the content is understandable, but needs revision to improve flow, eliminate redundancies, and ensure that the language meets international publication standards. Professional English editing is strongly recommended to refine grammar, sentence structure, and scientific style.
Response:
We thank the reviewer for this valuable suggestion.
A professional English editor make extensive revisions to our article, correcting grammar errors and improving sentence structure and streamlining the results and discussion. And we have expanded all the captions to make them easy to understand for all readers.

Reviewer 3 Report
Comments and Suggestions for Authors
This study compared the differences in gene expression, protein expression, and lipid composition between IMF and SCF in the longissimus thoracis et lumborum of fattening cattle using integrated transcriptomic, proteomic, and lipidomic approaches. The findings provide important insights into the mechanisms underlying IMF deposition and offer potential biomarkers and regulatory targets for improving beef quality.
Major comments:
-
Although lipidomics identified numerous differential lipids, the biological interpretation of these differences remains largely insufficient. There is a lack of in-depth functional annotation or metabolic pathway enrichment analysis. It is recommended to supplement relevant analyses to enhance the biological interpretation.
-
The authors only reported the total base yield of the sequencing results but did not specify key metrics, such as sequencing depth and read count. This omission makes it impossible to adequately assess whether the RNA-seq data meet the necessary standards for accurate gene expression quantification.
Minor comments:
-
The description of the experimental animals is insufficient. It would be better to supplement details such as age, rearing conditions, and diet.
-
The sample size meets the minimum requirement for analysis, but it may reduce the statistical power. If possible, expanding the sample size in future studies is recommended.
Author Response
Reviewer3:
This study compared the differences in gene expression, protein expression, and lipid composition between IMF and SCF in the longissimus thoracis et lumborum of fattening cattle using integrated transcriptomic, proteomic, and lipidomic approaches. The findings provide important insights into the mechanisms underlying IMF deposition and offer potential biomarkers and regulatory targets for improving beef quality.
Major comments:
- Although lipidomics identified numerous differential lipids, the biological interpretation of these differences remains largely insufficient. There is a lack of in-depth functional annotation or metabolic pathway enrichment analysis. It is recommended to supplement relevant analyses to enhance the biological interpretation.
Response:
Thank you for your positive comments and valuable suggestions to improve our manuscript.
In fact, 88 differential abundance lipids (DALs) were not enriched in any KEGG pathway in this study. Although we briefly described this result in line 262 (Section 3.4), it may not have been noticed due to a lack of necessary interpretation for this observation.
Following consultation with domain professors, we identified several potential reasons for the lack of KEGG pathway enrichment.
Firstly, different databases are used for identifying lipids and functional annotation. Lipid identification is primarily based on the LIPID MAPS database (https://www.lipidmaps.org/), which contains 49,907 lipid species. However, the KEGG pathway database, mainly used for functional analysis, currently annotates 698 lipids. This significant disparity in coverage likely led to the absence of KEGG pathways. Additionally, the molecular structures of lipids are complex, and their naming schemes are inconsistent. Certain lipids may be represented in the KEGG database as subclass categories rather than as individual compounds, which may further reduce annotation accuracy. In addition, we note that many published lipidomics studies (at end of the paragraph) have similar results, with limited KEGG pathway enrichment, implying a broader methodological challenge rather than a specific issue in our study.
To understand the functions of 88 differentially expressed lipids in our study, we searched them one by one in the NCBI PubMed database to match existing research. Three lipids were found: phosphatidylinositol (PI) (18:1/20:4), phosphatidylcholine (PC) (15:0/18:2), and lysophosphatidylethanolamine 18:0 (reference 50-52). Therefore, we mainly analyze the functions of these three substances and their subclasses (phosphatidylinositol and phosphatidylcholine) in lines 346-358 (Section 4.3). We have also revised the relevant section in the manuscript to make it easier to read and understand.
Published lipidomics studies with limited KEGG pathway.
Kyle, J.E.; Burnum-Johnson, K.E.; Wendler, J.P.; et al. Plasma lipidome reveals critical illness and recovery from human Ebola virus disease. Proc Natl Acad Sci U S A. 2019; 116(9):3919-3928. doi: 10.1073/pnas.1815356116.
Vriens K, Christen S, Parik S, et al. Evidence for an alternative fatty acid desaturation pathway increasing cancer plasticity. Nature. 2019; 566(7744):403-406. doi: 10.1038/s41586-019-0904-1.
Zhang, H.; Shao, X.; Zhao, H.; et al. Integration of Metabolomics and Lipidomics Reveals Metabolic Mechanisms of Triclosan-Induced Toxicity in Human Hepatocytes. Environ Sci Technol. 2019; 53(9):5406-5415. doi: 10.1021/acs.est.8b07281.
- The authors only reported the total base yield of the sequencing results but did not specify key metrics, such as sequencing depth and read count. This omission makes it impossible to adequately assess whether the RNA-seq data meet the necessary standards for accurate gene expression quantification.
Response:
We thank the reviewer for raising this important point.
We supplemented the sequencing depth in lines (Section 2.3), and added the total clean reads, mapping reads to genome, and unique reads in Section 3.1 and Table 1.
In lines 107-110: “Samples, with RNA integrity number (RIN) > 8.6 and the 28S to 18S rRNA ratio > 1.0, were conducted to RNA sequencing in an Illumina NovaSeq 6000 platform (Illumina, USA) at an approximately 6× sequencing depth.”
“3.1. Transcriptomic profilies and differentially expressed genes (DEGs)
Transcriptome sequencing generated 40.52 GB of clean data (SCF: 20.46 GB; IMF: 20.06 GB; Table 1), and 270 107 766 clean reads with 136 396 756 clean reads in SCF group and 133 711 010 in IMF. Q30 scores of all samples exceeded 90%, confirming data reliability. A total of 117 206 407 clean reads in SCF and 115 876 107 clean reads in IMF were mapping to the reference genome, respectively, with average mapping rates of 85.87% and 86.66% in SCF and IMF group. And the mapping ratio of unique reads to genome were 84.32% (SCF) and 84.97% (IMF).”
Minor comments:
- The description of the experimental animals is insufficient. It would be better to supplement details such as age, rearing conditions, and diet.
Response:
We thank the reviewer for this valuable suggestion.
We add a more detailed description of the experimental animals in lines (Section 2.2). The composition and nutrient content of the basal diets are shown in Supplementary S1.
2.2 Animals, Management and Sample Collection
Three thirty-month-old Angus steers (initial body weight of 573.6 ± 8.60 kg) were sourced from a commercial farm (36°24′N, 116°23′E) in Dezhou City of Shandong Province, and were fed on a basal diets with 5.41 MJ/kg net energy gain during a 240-days fattening period. The basal diet was designed in compliance with NRC (2016) guidelines to fulfil the nutrient requirements of beef cattle. The composition and nutrient content of the basal diet are shown in Supplementary S1. Throughout the experimental period, all cattle were housed in the same natural environment, and were provided twice daily feedings at 07:00 and 17:00 with ad libitum access to potable water.
- The sample size meets the minimum requirement for analysis, but it may reduce the statistical power. If possible, expanding the sample size in future studies is recommended.
Response:
We thank the reviewer for their insightful comment regarding the sample size.
We acknowledge that the sample size (n=3 steers) may indeed have reduced the statistical power for detecting more subtle effects. We have explicitly acknowledged this limitation in the revised Discussion section, and will expand the cohort in our future follow-up studies to improve the robustness and statistical power.

Reviewer 4 Report
Comments and Suggestions for Authors
Review
Review of the article “Comparative analyses of gene and protein expression, and the lipid contents in intramuscular and subcutaneous fat tissues in fattening steers”
Kaixi Ji 1, Ming, Yang 1,2, Ziying TAN 1,2, Hongbo ZHAO 1* and Xianglun Zhang
The purpose of the article was to elucidate the genetic mechanisms underlying IMF deposition by analyzing tissue-specific genes, proteins, and lipid metabolites of the IMF and SCF in cattle via integrated transcriptomics, proteomics, and lipidomics.
My comments:
- Simple Summary: Intramuscular fat (IMF) represents a critical economic trait……
Should be inserted (IMF)
- Сan be deciphered PI and PC (see line 32-33)
Introduction
The introduction provides a fairly comprehensive description of the data on fat in intramuscular fat and subcutaneous fat and the problems that the authors of the article solved.
- Materials and Methods
Materials and Methods are presented quite clearly.
- Results.
I have no comments in the results chapter.
- Discussion
The authors have done a lot of work to obtain the corresponding results. I offer some clarifications that may serve for further experiments
One of the functions of fat in animal muscles is as a potential source of energy. It may be that intensive fat accumulation occurs at a certain stage of the animal’s development and with increased (excessive) nutrition. You used Angus steers fed a basal diets with 5.41 91 MJ/kg. If you overfed the bulls, then maybe the genes that ensure the accumulation of fat in the muscles would be turned on. So you correctly note: “The inverse correlation between glycolysis and lipid metabolism in the IMF may reflect glucose diversion toward de novo lipogenesis”. (Line 337)
Since you used the small sample size, you reliably identified only genes with high expression and their corresponding proteins and lipids products. Since fat accumulation can occur slowly (and correspondingly moderate gene expression) as an animal ages, increasing the number of animals will allow you to notice this process.
Are there any published papers on the associa

Author Response
Reviewer4
Comments and Suggestions for Authors
Review
Review of the article “Comparative analyses of gene and protein expression, and the lipid contents in intramuscular and subcutaneous fat tissues in fattening steers”
Kaixi Ji 1, Ming, Yang 1,2, Ziying TAN 1,2, Hongbo ZHAO 1* and Xianglun Zhang
The purpose of the article was to elucidate the genetic mechanisms underlying IMF deposition by analyzing tissue-specific genes, proteins, and lipid metabolites of the IMF and SCF in cattle via integrated transcriptomics, proteomics, and lipidomics.
Response:
Thank you so much for your suggestions on improving the paper.
My comments:
Simple Summary: Intramuscular fat (IMF) represents a critical economic trait……
Should be inserted (IMF)
Response:
We revised the contents in line 11.
Can be deciphered PI and PC (see line 32-33)
Response:
We revised the content in lines 32-33:
“The abundances of both phosphatidylinositol (PI) (18:1/20:4) and phosphatidylcholine (PC) (15:0/18:2) were positively correlated with APOE.”
Introduction
The introduction provides a fairly comprehensive description of the data on fat in intramuscular fat and subcutaneous fat and the problems that the authors of the article solved.
Response:
Thank you for your positive assessment of the Introduction.
Materials and Methods
Materials and Methods are presented quite clearly.
Response:
Thank you for your positive assessment of the Materials and Methods
Results.
I have no comments in the results chapter.
Discussion
The authors have done a lot of work to obtain the corresponding results. I offer some clarifications that may serve for further experiments
One of the functions of fat in animal muscles is as a potential source of energy. It may be that intensive fat accumulation occurs at a certain stage of the animal’s development and with increased (excessive) nutrition. You used Angus steers fed a basal diets with 5.41 91 MJ/kg. If you overfed the bulls, then maybe the genes that ensure the accumulation of fat in the muscles would be turned on. So you correctly note: “The inverse correlation between glycolysis and lipid metabolism in the IMF may reflect glucose diversion toward de novo lipogenesis”. (Line 337)
Response:
Thank you very much for clarifying the underlying reasons for the inverse correlation between glycolysis and lipid metabolism in the intramuscular adipose tissue. This insightful comment is very helpful for our further experiments.
Since you used the small sample size, you reliably identified only genes with high expression and their corresponding proteins and lipids products. Since fat accumulation can occur slowly (and correspondingly moderate gene expression) as an animal ages, increasing the number of animals will allow you to notice this process.
Response:
Thank you for your insightful comment on the sample size and its impact on detecting moderately expressed genes.
We acknowledge that the sample size (n=3 steers) in the present study, while sufficient to identify genes with high expression levels and associated lipids, may indeed have limited the detection more subtle, yet biologically important, changes in moderately expressed genes. As Reviewer 3 noted, fat accumulation occurs slowly, and is mediated by genes with moderate expression levels. Larger sample sizes are required for identifying them.
We fully appreciate this constructive suggestion, and will prioritize larger sample sizes in a future study to comprehensively understand the underlying lipogenic mechanisms in intramuscular fat.
Are there any published papers on the association of genes (GWAS) with marbling in cattle and do they match the ones you found?
Response:
We searched for a large number of studies about the association of genes (GWAS) with marbling in cattle. FABP4 and SCD were associated with marbling score in Hanwoo cattle (Lee, HJ 2020; Lee, SH, 2010; 2008). The FABP4 g.3631 (G>A) allele was associated with a higher marbling score (Lee, 2020), and the CT genotype and T allele in SCD g. 10329 (C>T) were positively correlated with quality grades (Yu, et al. 2025). Regretfully, no results directly match the genes involved in calcium signaling or the glycolysis pathway, including ATP2A2, ENO3, FBP2, MYH7, MYL3, MYLK, PYGM, TNNC1, and TPM2. The single nucleotide polymorphism of other genes in the calcium signaling pathway affected meat quality. Calmodulin-dependent protein kinase II (rs452209056) was negatively associated with carcass weight in beef cattle (Wang, et al 2020). Additionally, PHKG1 (rs697732005) and ATP1A (rs344435545) affected the intramuscular fat contents in pigs (Wang, et al 2019). These results demonstrate that the calcium signaling pathway plays an important role in regulating marbling formation.
Reference
Lee, H.J.; Chung, Y.J.; Jang, S.; et al. Genome-wide identification of major genes and genomic prediction using high-density and text-mined gene-based SNP panels in Hanwoo (Korean cattle). PLoS One. 2020, 15(12):e0241848. doi: 10.1371/journal.pone.0241848.
Lee, S.H.; Cho, Y.M.; Lee, S.H.; et al. Identification of marbling-related candidate genes in M. longissimus dorsi of high- and low marbled Hanwoo (Korean Native Cattle) steers. BMB Rep. 2008; 41(12):846-51. doi: 10.5483/bmbrep.2008.41.12.846.
Lee, S.H.; van der Werf, J.H.; Lee, S.H.; et al. Genetic polymorphisms of the bovine fatty acid binding protein 4 gene are significantly associated with marbling and carcass weight in Hanwoo (Korean Cattle). Anim Genet. 2010; 41(4):442-4. doi: 10.1111/j.1365-2052.2010.02024.x. Epub 2010 Mar 15.
Wang, Y.; Zhang, F.; Mukiibi, R.; et al. Genetic architecture of quantitative traits in beef cattle revealed by genome wide association studies of imputed whole genome sequence variants: II: carcass merit traits. BMC Genomics. 2020; 21(1):38. doi: 10.1186/s12864-019-6273-1.
Wang, B.; Li, P.; Zhou, W.; et al. Association of Twelve Candidate Gene Polymorphisms with the Intramuscular Fat Content and Average Backfat Thickness of Chinese Suhuai Pigs. Animals (Basel). 2019; 9(11):858. doi: 10.3390/ani9110858.
Yu, J.; Naseem, S.; Park, S., Hur S, et al. Gene Polymorphism and Association with Carcass Traits and Fatty Acid Profile in Hanwoo Cattle. Animals (Basel). 2025; 15(6):897. doi: 10.3390/ani15060897.
